# A Theoretical Framework and Conceptual Design for Engaging Children in Therapy at Home—The Design of a Wearable Breathing Trainer

**DOI:** 10.3390/jpm9020027

**Published:** 2019-05-20

**Authors:** Lara Siering, Geke D.S. Ludden, Angelika Mader, Hellen van Rees

**Affiliations:** 1Interaction Design, Faculty of Engineering Technology, University of Twente, 7522 NB Enschede, The Netherlands; l.siering@student.utwente.nl; 2Human Media Interaction, Faculty of Electrical Engineering, Mathematics and Computer Science, University of Twente, 7522 NB Enschede, The Netherlands; a.h.mader@utwente.nl; 3Hellen van Rees, 7521 BE Enschede, The Netherlands; hellenvanrees@gmail.com; 4Smart Functional Materials, Saxion University of Applied Sciences, 7511 JL Enschede, The Netherlands

**Keywords:** design for engagement, wearable technology, digital health, interaction design, sensory pleasure, children

## Abstract

Wearable technologies are being implemented in the health and medical context with increasing frequency. Such technologies offer valuable opportunities to stimulate self-management in these domains. In this context, engagement plays a crucial role. An engaged patient is a patient who is emotionally involved and committed to the therapy or care process. Particularly for children who have to follow some sort of therapy, engagement is important to ensure a successful outcome of the therapy. To design for engagement, a framework based on theories of motivation in child therapy was developed. This framework was applied to the design of a wearable breathing trainer for children with asthma and dysfunctional breathing. As such, the present paper provides knowledge about the implementation of theory on engagement and motivation in design. Expert and first user evaluations found that the resulting prototype is appealing, perceived as useful, and may engage children in breathing training and stimulate self-management.

## 1. Introduction

Wearable technologies increasingly draw attention from researchers and industry in the healthcare domain. Although currently the focus is mainly on sensors measuring physiological parameters to study data and to provide insights into the state and behavior of the user [1,2,3,4,5], wearable devices are also suitable to promote self-management and therapy at home. Well-known wearable devices that support self-management are, for example, fitness trackers or smartwatches that were developed to provide support in increasing physical activity. Reasonably novel directions in the field of wearable technology are developments in smart textiles or smart clothing. Smart clothing has been defined as a meaningful and useful integration and/or adaptation of electronic systems with intelligent functions in textile and non-textile clothing [6]. Examples of such smart clothing are the Solar Dress by Paulien van Dongen [7], the closed loop smart athleisure fashion by Marina Toeters [8], and the Smart Rehabilitation Garment [9], indicating a wide applicability. The main application of smart clothing is sensor-based monitoring, such as the acquisition of vital signs in medical surveillance or the estimation of physical activity in sports [10]. With its unobtrusive character, smart clothing provides a convenient integration with the everyday life of the wearer. Next to sensing, smart clothing finds application in the field of human–technology interactions. Current interaction techniques on mobile devices make use of explicit input from users via touch and speech and output via sound and visual information on a screen. Smart clothing offers the additional possibility of an intimate body contact and pioneers novel forms of output in the form of tangible and embodied interactions, such as haptic or shape-changing feedback [11,12]. This combination of, on the one hand, sensing and monitoring technologies and tangible and embodied interaction on the other, makes smart clothing suitable for home therapy and self-management. In this paper, we discuss this new field of application in the case of smart clothing for children with dysfunctional breathing.

Chronic respiratory disorders such as asthma and dysfunctional breathing (DB) are common in childhood and imitate or mask each other, as they have similar symptoms and can occur in the same child. DB is defined as chronic or recurrent changes in the breathing pattern, causing respiratory and non-respiratory complaints [13]. These changes are commonly shown as excessive upper chest and accessory muscle activity with little changes in thoracic volume [14]. Its symptoms are dyspnea with normal pulmonary function, deep sighing, chest pain, chest tightness, frequent yawing, hyperventilation, and breathlessness during exercise [15]. These symptoms have a wide range of severity from mild and hardly obtrusive, to severe hyperventilation attacks and panic [15]. Children with respiratory disorders are frequently referred to a physical therapist, who educates children in self-assessment and works on improvement of breathing technique. In this context, children are asked to perform further breathing retraining exercises at home. However, these exercises are often not engaging, and children are not likely to be self-motivated to do these exercises, whereas engagement and motivation to sustain an active involvement is key to a successful outcome of therapy. In addition, there is a lack of tools that can guide and support caretakers and healthcare professionals to monitor the progress of skills, such as self-assessment and breathing technique in children. Here, smart clothing could offer a solution to, on the one hand, supporting self-management and engagement and offering a possibility for monitoring on the other. A wearable breathing trainer has the potential to increase the training frequency at home and therefore decrease contact moments with the physiotherapist. By providing feedback and motivation, the breathing trainer could help the child to gain the skills to cope with asthma or solve the problem of dysfunctional breathing.

This paper will, through the case of presenting the design methodology of a wearable breathing trainer, show how theory and frameworks on engagement and motivation for child therapy can be translated into design of wearable technology to enhance engagement in at-home training and autonomy of the patient.

## 2. Design Methodology

Engaging patients in therapy and motivating them to comply and sustain an active involvement is the key to a successful outcome of a therapy. Motivation and engagement have been described as essential aspects of therapy and rehabilitation [16], in particular when supporting self-management of the patient. Therefore, it is crucial to understand human motivation and engagement for the design of wearable devices that support self-management and behavior change. We therefore started the design process by studying how motivation and engagement of children are understood and what factors of child development should be considered. The insights from theory on motivation and engagement and from developmental psychology were combined in a theoretical framework. In the next step, we applied the framework and the resulting design requirements to the design of a wearable breathing trainer to be used for physical therapy of children with dysfunctional breathing (see Figure 1). We focused here on children aged from 6 to 12 years as they are the main age group affected by dysfunctional breathing. This age group is defined as middle childhood according to developmental psychology [17].

### 2.1. Theoretical Framework for Engaging Children in Therapy at Home

Authors in different domains, including digital health informatics, have focused on understanding human engagement [18,19]. These studies present mostly coherent and clear descriptions of possible reasons for engagement and mechanics that make people engaged. Many models or theories on engagement take the Self-Determination Theory (SDT) as a basis. For example, SDT has been used to explain motivational aspects in gaming and education [20,21,22,23]. A model which applies the SDT in child therapy is the SCOPE-IT model [24]. This model was developed for the context of human therapists for occupational child therapy, but it is also applicable in other fields. With this design case, we want to scrutinize if the theoretical background can support the design decisions for wearable devices to inform a design that aims at sustaining intrinsic motivation. The SCOPE-IT model offers a means of understanding why a child is engaged, or becomes disengaged, and provides directions about how to facilitate motivation, build self-determined action, and promote engagement.

The authors of the SCOPE-IT model [24] developed the Rocket Model of Motivation, which helps to understand the motivational forces supporting successful therapy. It metaphorically compares motivation and engagement to a rocket launch. The different motivational stages of self-determination and their external and internal regulatory styles offer a foundation for design decisions supporting engagement. External regulations are understood as motivational influences from the outside, whereas internal regulations derive from the child itself. The authors of [24] offer useful suggestions to enhance intrinsic motivation in different stages that can be implemented in the design of wearable devices. When a child is intrinsically motivated, thus motivated from inside, the child is more likely to engage and stay engaged in therapy. According to the authors of [24], intrinsic motivation can be divided into three types: intrinsic motivation to know, to experience mastery or competence, and to experience stimulation either from sensory pleasure or through flow. In the context of wearable devices/smart clothing, the stimulation from sensory pleasure offers possibilities because wearable devices offer multiple ways to stimulate the senses of the wearer, as we have also stated in the introduction.

The following regulatory elements of the Rocket Model of Motivation are considered useful in the context of the design of wearable devices/smart clothing to support intrinsic motivation through the fulfilment of the three psychological needs of the SDT.
Providing (trustworthy) informational feedbackOffering personal goal settingProviding (personal) choicesSmall steps towards skill acquisitionLetting the child make decisions

Next to the fulfilment of the psychological needs, the child-centered factors also play a role in motivating children [24]. These factors can be related to the children’s abilities, interests, and developmental needs, which highly influence the design of technology for children [25,26]. Theories on cognitive development constitute a rich knowledge base for designers who are designing technologies for children. Cognitive development, perception, attention, memory, and problem-solving play central roles in designing child–technology interactions. After analyzing the different theories on child development [17,25,26], we determined the following middle childhood abilities in perception, attention, memory, and problem solving as relevant for the design of the breathing trainer. The design should
Focus on a few closely related tasksMeet the memory capacity (5–6 chunks of information)Support memory strategiesSupport goal settingUse quantitative measures for the provision of feedback

The requirements emanating from principles of engaging children in therapy and from the abilities of children prompted us to make the link to game mechanics due to their analogy. The use of game mechanics is already described as a possible way to motivate and engage children in the field of physical therapy [27]. Game mechanics aim to make activities more game-like by using game design characteristics and elements [28]. Typical elements are, for instance, points, badges, leader boards, meaningful stories, avatars, or teammates [29]. A meaningful story and avatars are likely to support the need for relatedness and to enhance the engaging factor of wearable devices.

Noteworthy are the sensory stimulation, the experience of competence, and the meaningful story as part of intrinsic motivation. Pleasurable sensory stimulation can be easily implemented in wearable devices and especially in smart clothing because it is in direct contact with the body. It can then also serve as an intimate and direct channel of interaction that decreases cognitive load. Such new types of interaction, such as haptic feedback, open up new possibilities for emotional engagement through interaction with the device.

We combined all the elements that we used to inform our design into a framework. This framework guided the design of a wearable device for engaging children in therapy at home. Figure 2 shows the Design for Child Engagement (DCE) Framework.

We propose a theoretical framework in which the regulatory styles of the Rocket Motivation Model (Poulsen) acts together with child-centered factors as functional requirements to influence the design of a wearable device for motivation and engagement. The experiential requirements of a meaningful story, sensory stimulation, and experience of competence in combination directly influence the design decisions. All requirements together are expected to positively influence intrinsic motivation and engagement, which will lead to the desired outcome of behavior change and physiological effects. In this case, the behavioral change is that the child becomes motivated and regularly performs the exercises at home. The physiological effect will be a normal breathing technique.

### 2.2. Applying the Framework to Design a Wearable Breathing Trainer

The DCE Framework in Figure 1 was applied to the design of a wearable breathing trainer with the aim to engage children aged 6–12 years in breathing retraining sessions at home. Physiotherapists often practice breathing exercises with children with dysfunctional breathing patterns and suggest to them to practice at home by themselves daily. However, children often do not perform these exercises at all or not on a daily basis. At the same time, there is a lack of reliable feedback of progress to the physiotherapist. A smart garment could provide the necessary support and motivation to actually perform exercises on a daily basis while providing feedback about performance and progress. Because haptic feedback is an essential part in breathing exercises, the smart garment is particularly suitable to provide this. In addition, it can provide sensory stimulation which is part of intrinsic motivation [24].

To deepen the understanding of how the design of a wearable trainer can support engagement, we will specifically focus our discussion on the integration of motivational and engaging elements in design. We will illustrate how our design process resulted in a conceptual design of a wearable breathing trainer. Eventually, experiential prototypes of the conceptual design were developed and used for a first formative evaluation with medical experts and end users. For the design process, we describe the dominant design decisions with relation to the Design for Engagement Framework.

#### 2.2.1. Design

Based on the DCE Framework, we decided to combine a smart vest with a mobile application for the design of a wearable breathing trainer. This combination offered the best possibilities to implement the strategies defined to support the process of engagement. The implementation of all elements into a vest would have caused an overloaded experience during training and might have distracted from the breathing exercises. A mobile application offered a means to support the therapy not only during the actual training, but also before and after, in anticipation and reflection. During a training session, the vest will support the child in the exercises by providing feedback. After finishing the training, the child will connect the vest with the mobile application to see his or her overall progress. The mobile application will incorporate additional game elements to sustain the motivation of the child. The implementation of both functional and experiential requirements is outlined in detail in this section.

*Both haptic and visual feedback elements* are implemented directly in the vest. These sensory feedback types support the child during training by providing a guiding feedback and *feedback on performance*. The haptic feedback is a translation from the traditional therapy sessions where a therapist supports the child with the hands. With the hands positioned on the chest and abdomen, the light pressure on the chest and the drawback at the abdomen stimulate the child to breath using the abdominal area during inhalation. During exhalation, a light pressure on the abdomen can support the child. This type of guiding feedback provides sensory and perceptual stimuli, reflecting the performance itself [30]. To realize a similar haptic feedback, we applied the shape-changing textile ‘Textile Reflexes’ designed by Hellen van Rees (see Figure 3).

The Textile Reflexes material is made of separate squares. The square shape of the elements is optional, experiments were also done with more curved shapes, a combination of circles and ‘star’ shapes. The squares were finally chosen because they provide the most stable and strong material due to their equal, 90° angles. The squares are stitched together with a string across the diagonals, connecting each square to the next at the corners. This ensures that the corners remain a flexible point allowing the textile to grow, shrink, fold, and bend. No matter how big the basic squares are and how many of them are there, if the closed textile has a breadth 𝓁, the opened textile has a breadth 1.4 × 𝓁. The flexibility allows it to respond to the shape of the human body [31]. The shape-changing textile is made of recycled textile-waste felt and is a kinetic system. By adding actuators, it can become a robotic textile that is able to provide very different haptic experiences. A shape-changing robotic textile could be compared to a soft muscle that can apply or release pressure [31]. A guiding feedback provided by the robotic textile that generates more or less pressure by moving the squares is ideal to navigate the breathing of the child. Situated at the front of the breathing trainer, the robotic textile can expand and contract guiding the breath to the abdominal area. Haptic feedback in this form applied in the right position, has a great potential in a training context because it provides direct physical input on the body, thereby possibly lowering the cognitive load needed to respond to the feedback. Moreover, haptic feedback is not only another way of perceiving information, it also includes another experiential dimension. Haptic signals on the body, such as pressure or embracement, are attributed with meaning, which also become apparent from language. We use phrases like “a pat on the shoulder” to indicate approval or “hold me tight” to indicate a feeling of safety [32].

Visual feedback in the form of light is used to provide the child with information on how he or she is performing with regards to the position and rhythm of each set of breaths. Additionally, after the training, the visual feedback can be used to indicate the overall performance of a training session. In this way, we provide the child with informational feedback to support the need for competence and relatedness. To distinguish this feedback from the guiding haptic feedback, a different type of sensory feedback was selected. Because children of this age group prefer quantitative measures for feedback, we implemented the quantity aspect in the visual feedback. The light is implemented in the squares of the robotic textile with color-changing LEDs (see Figure 4).

Such a matrix of LEDs offers an easy implementation to provide clear feedback on position and rhythm of the child’s breath. The following feedback mode was developed: If the child performs a desired deep breath, all rows of the matrix light up. If the breath was not deep enough, just the first row lights up. The number of rows that light up depends on the accuracy of the position and increases with the accuracy. The feedback on the rhythm is visualized with the color of the LEDs: If the rhythm is too fast, the LEDs stay white; if the rhythm is good, the LEDs light up in a color that has previously been selected by the child.

The remaining requirements *feedback on progress*, *meaningful story*, *clear goal*, *challenges,* and *provision of choices* focus on sustaining continuous use and are implemented in the mobile application.

The use of two different devices, a smart vest and a mobile application, require a strong and meaningful linkage in design to create a unified whole. The link between the vest and the application is based on the light element. On the vest, the light visualizes the *feedback on performance* that will be translated into *feedback on the overall progress* in the mobile application. To make this linkage meaningful, the light is used within the story to indicate progress. For the transfer of light from vest to mobile application, the light is collected with the smartphone. After the training, the child picks up the smartphone and moves the phone over the light. An integrated Near Field Communication (NFC) chip detects the NFC compatible smartphone and turns off the light on the vest. At the same time, data is exchanged, and the application displays the light that was ‘collected’ by the child.

The game element of a *meaningful story* supports the linkage between the vest and the application. A story is a quite common game element and it can also support the need for relatedness. It creates the feeling of being part of something. Furthermore, a story can coordinate all the other game elements, vest and application, to form a common entity. For the conceptual design, we selected the theme of being on a mission through space, travelling from planet to planet to get back to Earth. To get to the next planet, the child has to collect a certain amount of light during the training. For the conceptual design, we developed two characters: a little space monster and a little astronaut. With these characters, the child can form a bond for this mission, creating a feeling of *we* instead of me.

Along with the story, the game elements of *goal setting* and *challenges* are implemented in the application. The goal to breathe normally is paralleled with getting back to Earth. The journey from planet to planet corresponds to the different breathing exercises children have to perform. Here, challenges and increasing difficulty support the need for competence. These challenges should be related to the different exercises of the breathing retraining.

The general goal of breathing in a better way and getting back to Earth is supplemented by a personal goal, such as being able to ride to school by bike or play a whole football match after the mission is completed (at the end of the therapy). By letting the child set its own goal, the element of *providing choices* is implemented. The personal goal could be set by the child, for example, in the onboarding process of the application. To ensure an attainable goal, the goal setting should be guided by the physiotherapist. In addition, the child is able to choose the color of the light during the onboarding process. By offering these choices, the system supports the need for autonomy.

#### 2.2.2. Prototyping

We created a first set of prototypes consisting of a wearable breathing trainer vest that uses the robotic textile in the front, an LED-matrix implemented in a separate matrix with the same structure as the robotic textile, and a simple click-through prototype of the application.

In the development of the vest we kept in mind the usual important aspects for creating a garment as well as additional aspects following the theoretical framework, in other words, designing for motivation of this specific target group. The usual aspects are, for example, ensuring reproducibility by drafting base patterns and using readily available confection techniques and textiles. After a draft version, we decided to treat the sizing somewhat differently. By creating a system of adjustable Velcro straps the vest can be used for a range of sizes, rather than just one specific size, making it easier to test and use with a target group that grows rapidly. This was particularly important because for the haptic feedback to have its desired effect, the vest needs to be close around the body.

Following the theoretical framework, for this application we needed to integrate the LED light matrix and add the mobile application to be able to set goals and connect the breathing exercises to a game. The textile reflexes panel was therefore developed with a circular hollow space in the middle of each square to hold an LED light. The size of the squares was adjusted to this (41 × 41mm) and cut by laser cutting. The Textile Reflexes panel was placed on the front and was made large enough to be able to cover the chest and the lower abdomen. With these considerations the panel ended up being 5 × 6 squares that are stitched to the base vest only on the top, left, and right rows, allowing the squares to move freely in between and have as much flexibility as possible (within the constraints of the pattern), and with that, more effective haptic feedback. In the case of this vest, that was a maximum contraction of 82 mm (closed the squares are 5 × 41 = 205 mm wide, open they are 205 × 1.4 = 287 mm). Colors and fabric for the vest were chosen to be gender neutral, but not white, to avoid the look of a medical product.

For the first evaluation we used a prototype of the wearable breathing trainer vest made by designer Hellen van Rees (see Figure 5 on the left). The design of the vest was inspired by the space theme to make the linkage between vest and mobile application even stronger. This prototype did not have electronic actuators but was operated manually. For this we integrated strings in the vest at the chest and the abdomen that, once pulled, contracted the vest for haptic feedback.

The LED-matrix (see Figure 5 on the right) was controlled by an Arduino microcontroller [33] in combination with an NFC shield to simulate the interaction with an Android smartphone. Via the NFC signal, the light simulation could be started and stopped. In this context, a simple animation was programmed that lights up the rows one by one until the whole matrix lights up.

The prototype of the mobile application consisted of a series of screens (a selection is displayed in Figure 6). Within the onboarding process (first time use of the application), the child is asked to set a personal goal, and the application teaches the child how the interaction with the vest works. A long, scrollable home screen illustrates the course of the mission, starting with the preparation in the rocket (cockpit) and ending back on Earth. To get from theme to theme, the child has to collect light during the training. This will fill the two bars, illustrating the tank levels of fuel (position of breath) and oxygen (rhythm of breath).

## 3. Formative Evaluation of the Design

We aimed the first user evaluation at acceptability and usability of the wearable breathing trainer system for health professionals and children and its expected influence engagement with the therapy. At the same time, the evaluation was set up in such a way that it would provide us with useful feedback from different stakeholders on the design implementation. We therefore applied a formative evaluation methodology from the field of human–computer Interaction, as this is more suited for this conceptual design case than a randomized controlled trial [34]. We presented the conceptual design to a child physiotherapist specialized in breathing retraining and a child pulmonic specialist. Furthermore, we presented the different elements of the breathing trainer to ten children aged 6–12 years to assess if they understood the system, were attracted by it, and would be motivated to use it.

### 3.1. Evaluation by Experts

The prototype of the conceptual design of the breathing trainer was evaluated with a child physiotherapist from the physiotherapy practice t’Dijkhuis in Borne. The physiotherapist stated that the concept of the breathing trainer is an autonomous system that acts like a therapist at home. In her view, the vest with its haptic feedback is very suitable to guide and support the child during the breathing retraining at home. Furthermore, the physiotherapist commented that the use of the light should be limited to just giving feedback after a series of breaths. Otherwise, the child will be distracted and will keep looking down on the abdomen which would hinder physiological breathing. Regarding the mobile application, the physiotherapist suggested dividing the target group into different age-subgroups. At the age of six, children will probably experience the application and the interaction as challenging by itself, whereas older children probably need more interactions and challenges within the application. However, the physiotherapist sees the use of a supportive mobile application as an asset.

According to the physiotherapist, the combination of selected motivational elements could support a sustained use of the breathing trainer and frequent exercising at home. In this context, the physiotherapist highlights the implementation of goal setting and the challenge children will feel to reach this goal. The physiotherapist sees in the use of such a breathing trainer, the advantage of being able to reduce the contact moments in therapy for children.

As pediatricians diagnose dysfunctional breathing in children and refer them to a physiotherapist, we also presented the conceptual design to a child pulmonic specialist of Medisch Spectrum Twente (MST), the hospital of Enschede. The pediatrician was positive about the design implementation and highlighted the motivational aspect of the breathing trainer. In contrast to the physiotherapist, the pediatrician is more interested in monitoring the dysfunctional breathing of the child and sees possibilities to add the monitoring task to the breathing trainer. In his view, the system should be able to both detect and analyze respiratory disorders and provide real-time feedback. This gives children, parents, and health care providers valuable information regarding the origin and severity of symptoms that could lead to improved self-management.

### 3.2. Evaluation by the Target Group

The concept of the wearable breathing trainer was presented to ten children (aged 6–13 years, 6 girls) in physiotherapy practice t’Dijkhuis in Borne to document first impressions of the end user group. The protocol for the study was presented to the medical ethical committee of the MST hospital of Enschede, the committee decided that a complete ethical approval was not necessary for this formative evaluation. Parents of participants were informed before the study date and were asked to sign a consent form. All participants were involved in an exercise program for children with chronic or long-term diseases. They were not necessarily affected by respiratory disorders. However, all were familiar with medical treatment and physiotherapy.

For the formative evaluation, the prototype of the vest in combination with the conceptual LED-matrix and the prototype of the mobile application were shown. Feedback from the target group was gathered with an unstructured qualitative interview supported with the laddering method with specified ‘what’ questions [35]. To have the participants reflect on how they felt about certain aspects of the conceptual design, the LEMtool, that depicts a range of eight emotions (joy, sadness, desire, disgust, fascination, boredom, satisfaction, dissatisfaction, see Figure 7) was adapted for the interview [36]. With this tool, the researcher was able to trigger responses towards aesthetics, usability, and likability of the prototypes.

In general, the concept was received positively. Nine out of ten children indicated that they felt positive about the concept of the breathing trainer (pointing at positive emotions: 1× joy, 1× desire, 3× fascination, 4× satisfaction). One 6-year old girl indicated she felt a negative emotion (disgust). Regarding the appearance of the vest, all children asked for the use of more and brighter colors: “*Personally, I like things with many colors and gray is a little bit boring. Now I wear red, but I like all colors*” (10-year old girl). Some children indicated that the squares should be hidden to make it look like a normal shirt: “*When I see something so different, I often think that I do not have the courage to put this on*” (7-year old girl), “*For me, it looks unfinished. I would put another layer on top of the squares*” (12-year old boy). Only one girl explicitly mentioned that she likes the appearance of the squares. The 8-year old boy tested the vest and the haptic feedback, and he liked it. He was able to feel the pressure, but it was noted that the activation point on the abdomen should be positioned higher. The concept of the light as feedback was understood very well, all children correctly noted that more lights stand for a good performance. In addition, the possibility to choose their favorite color was received very positively. The children liked the application as a supporting tool. However, they wished for more interaction: “*I would like to have something like a small game that I can play after my training* (12-year old boy). Younger children mentioned that they would prefer a different story, or different characters: “*Many children like animals, maybe we can make it a zoo story*” (10-year old girl), “*I like firemen, so I would like to help a fireman”* (7-year old girl). Nevertheless, for all children, the interaction between vest and application was clear and they were positive and even in awe about it.

## 4. Discussion

We have presented the development of a framework that combines existing theories and translates them to a design implementation. The framework proved useful to inform our design process to design a wearable device in mainly two ways. First, it provided inspiration as a starting point for design, second, it provided a solid ground to make design decisions. Some important decisions made based on the DCE Framework during the design process were later confirmed to match the target groups needs in the evaluation. For example, the decision to use light in the form of an LED matrix with a focus on the quantitative aspect as feedback on performance was evaluated as markedly explicit and understandable. Another important design decision that was made based on the framework was to combine the vest with a mobile application. With using the data on performance from the vest to increase the challenge, it is guaranteed that the child will stay engaged during the therapy.

The formative evaluation with children was used to generate a first impression on how the target group receives the concept, to assess whether they would initially be motivated to use the breathing trainer for therapy at home and to gather feedback for further improvement of the design. The evaluation indicated that the target group liked and could use the system that was designed. Nearly all of the participates pointed out that they are likely to use the trainer. However, the appearance of the vest has to be changed to make it more appealing for the target group. Experts pointed out that the current system would be highly usable to treat dysfunctional breathing by motivating children to train with this wearable device. In addition, it was pointed out that the wearable device may also offer a means to analyze the origin and severity of symptoms. Although this addition would make the system much more complex it would also make it more complete. Adding this link to the diagnostic phase could offer a means for further personalization of the system that is not only based on a child’s preferences but also on his or her personal condition and progress.

The design approach of the DCE framework makes explicit how to design for engagement and motivation based on acknowledged theories. The framework combines the design-relevant elements of motivational and developmental theory. The work presented in this paper can therefore be seen as an example of how available theory on engagement and motivation and on child development and therapy can be used in the design of innovative medical devices. This type of work is needed to create a knowledge base on how to design engaging interactions with wearable and other interactive devices for (child) therapy in the home context.

## Figures and Tables

**Figure 1 jpm-09-00027-f001:**
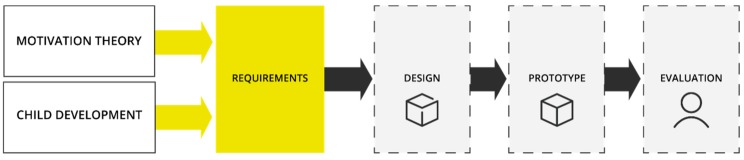
Design process based on the theory on motivation and child engagement and child developmental psychology.

**Figure 2 jpm-09-00027-f002:**
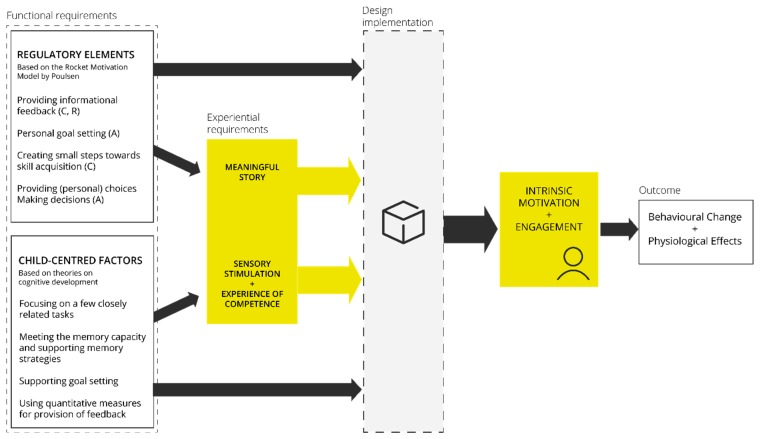
The Design for Child Engagement (DCE) Framework. Functional and experiential requirements determine the design of engaging wearable devices. The functional requirements are derived from the regulatory elements of the Rocket Motivation Model (Poulsen) and from the child-centered factors based on child developmental psychology. When implemented in the design, the experiential requirements together with the functional requirements facilitate intrinsic motivation and engagement. Eventually, the wearable device leads to behavioral change and physiological effects. In this case, the behavioral change is that the child will regularly perform the exercises at home and the physiological effect will be an improved breathing technique.

**Figure 3 jpm-09-00027-f003:**
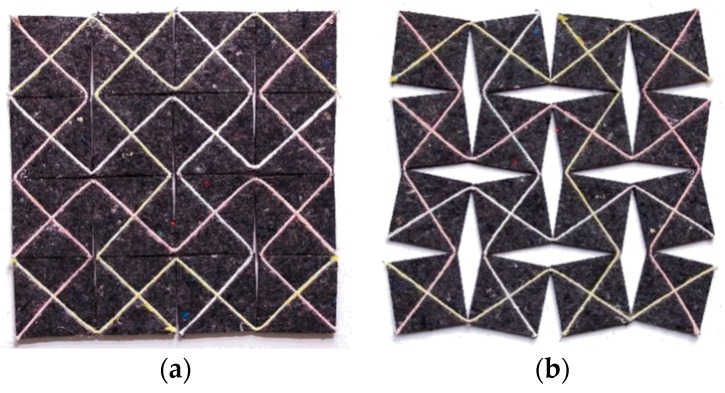
The moving squares of the textile Textile Reflexes in closed (**a**) and open (**b**) position.

**Figure 4 jpm-09-00027-f004:**
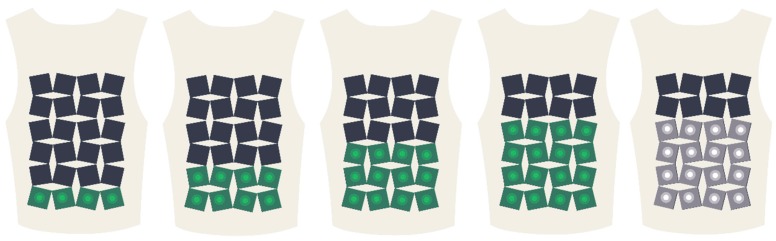
The LED-matrix showing an increase in the lighted rows that indicate the feedback on performance regarding the position of the breath. The white light informs the child that the rhythm was not good.

**Figure 5 jpm-09-00027-f005:**
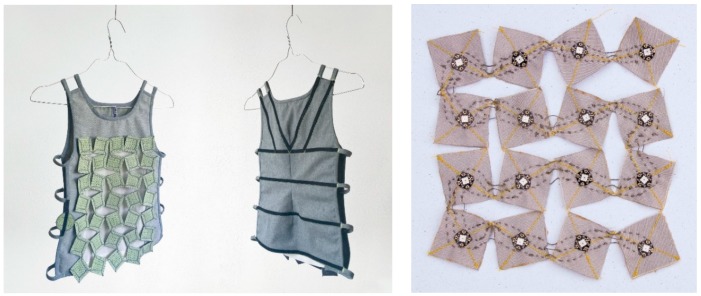
First prototype of the Wearable Breathing Trainer (WBT) with the square-shaped robotic textile and the LED-matrix.

**Figure 6 jpm-09-00027-f006:**
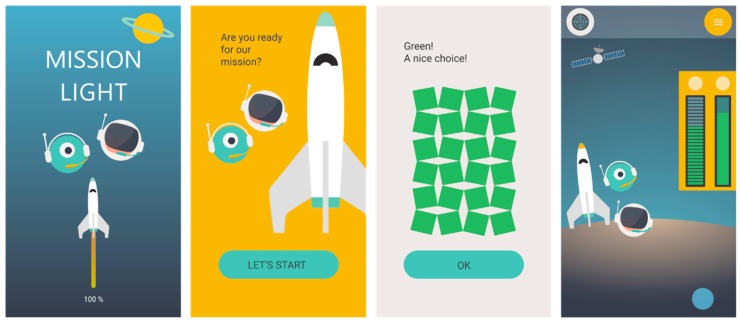
Screens of the click-through prototype application displaying the start screen (left), introduction and onboarding process (middle two) and home screen (right).

**Figure 7 jpm-09-00027-f007:**
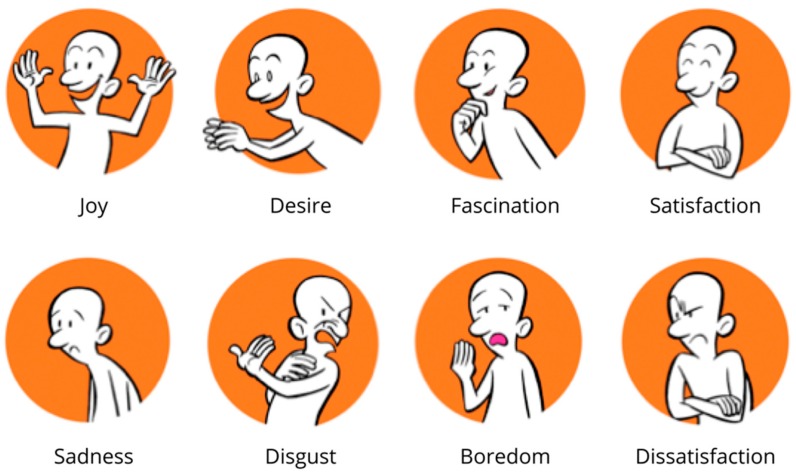
The LEMtool cartoon character and the expressed eight emotions. Adapted from [36].

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
