# Peer review of "A Theoretical Framework and Conceptual Design for Engaging Children in Therapy at Home—The Design of a Wearable Breathing Trainer"

_jpm, 2019, doi:10.3390/jpm9020027_

Round 1
Reviewer 1 Report
- Some sentence/word choices and grammar require review/edits
- Clearly outline why you are designing this wearable/smart clothing for this specific patinet population. What are you trying to achieve (e.g. specific outcome improvements? what outcomes?) It sounds like your theory is that by improved engagement you'll get a better result. Write plainly so your purpose and plan is clear.
- Clarify meaning of "field of interaction"- Reword this in plain language - most readers won't be certain of the meaning of this: This paper will, through the case of presenting the design methodology of a wearable breathing trainer, show how theory and frameworks on engagement and motivation for child therapy can be 65 translated into design of wearable technology.
- Results and discussion could be reduced and length and focus more on why our new approach was needed and exactly why its better and in what situations. Since your framework isn't put to the test, what exactly should the reader take away? When should it be deployed and when not?
- Consider summarizing key takeaways and findings
Author Response
We thank the reviewer for the thoughtful comments about our work and have used them to further improve the paper. Please find below a point-by-point response to the comments that were raised.
Reviewer comment:Some sentence/word choices and grammar require review/edits
Response:We have reviewed the paper on grammar, spelling and punctuations, edits were made throughout as is visible from the track-changes in the revised manuscript.
Reviewer comment:Clearly outline why you are designing this wearable/smart clothing for this specific patient population. What are you trying to achieve (e.g. specific outcome improvements? what outcomes?) It sounds like your theory is that by improved engagement you'll get a better result. Write plainly so your purpose and plan is clear.
Response:We have highlighted in line 60-72 that engagement and motivation are the key to a successful outcome of therapy;
„A wearable breathing trainer could increase the training frequency at home and therefore decrease contact moments with the physiotherapist. By providing feedback and motivation, such a breathing trainer could help the child to gain the skills to cope with asthma or solve the problem of dysfunctional breathing.“
“This paper will, through the case of presenting the design methodology of a wearable breathing trainer, show how theory and frameworks on engagement and motivation for child therapy can be translated into design of wearable technology to enhance engagement in at-home-training.”
Reviewer comment:Clarify meaning of "field of interaction"- Reword this in plain language - most readers won't be certain of the meaning of this: This paper will, through the case of presenting the design methodology of a wearable breathing trainer, show how theory and frameworks on engagement and motivation for child therapy can be 65 translated into design of wearable technology.
Response:We have reworded this in line 42 to „human-technology-interaction“ (an explanation is given in the following sentences)
Reviewer comment:Results and discussion could be reduced and length and focus more on why your new approach was needed and exactly why its better and in what situations. Since your framework isn't put to the test, what exactly should the reader take away? When should it be deployed and when not?
Response:From line 320 on we have shortened the evaluation section (see track-changes)
In line 440-443 we have added what we see as the most important take away and how we see others can build on this work.
“The design approach of the DCE framework makes explicit how to design for engagement and motivation based on acknowledged theories. The framework combines the design-relevant elements of motivational and developmental theory. The work presented in this paper can therefore be seen as an example of how available theory on engagement and motivation and on child development and therapy can be used in the design of innovative medical devices. This type of work is needed to create a knowledge base on how to design engaging interactions with wearable and other interactive devices for (child) therapy in the home context.”
Reviewer comment:Consider summarizing key takeaways and findings
Response: We have summarized the main contribution of the paper as an additional final paragraph (see previous response).
Reviewer 2 Report
Dear Authors,
This is an interesting manuscript on the development of a wearable device for breathing training. The novelty and quality of this research is appealing since it addresses the realization of a wearable device that might allow improving rehabilitation strategies in children affected by dysfunctional breathing using an interactive approach. The paper can be published after minor revisions.
The following comments need to be addressed prior to publication.
General Overall Comment 1: You should provide more details that better describe the wearable device. How many T-shirts did you realize? Was there any variability in the behavior of different t-shirts that account for the reproducibility of the device here described? Can the production of these wearable devices be scaled up? You need to better address all these points within the manuscript.
General Overall Comment 2: How many are the squares stitched on the t-shirt? Would a different shape (e.g., a rectangle, a circle, etc.) affect the performance of the device? Why did you choose the square shape for the sensing elements? You need to better clarify how form (size, shape) and function (pressure applied) correlate together. You need to better justify the reason why this is the final design.
Comment 3: Page 1, Line 29. You need to add few more references like some reviews that focus on the non-invasive monitoring of physiological parameters (e.g., in saliva, sweat, breath, etc.) using wearable devices.
Figure 2: The figure needs to have a higher resolution. The text in the boxes on the left hand side is not clear enough. You might also need to increase the size of the font you used.
Author Response
We thank the reviewer for the kind words and the thoughtful comments about our work and have used them to further improve the paper. Please find below a point-by-point response to the comments that were raised.
Reviewer Comment:You should provide more details that better describe the wearable device. How many T-shirts did you realize? Was there any variability in the behavior of different t-shirts that account for the reproducibility of the device here described? Can the production of these wearable devices be scaled up? You need to better address all these points within the manuscript.
Response:We added process and considerations of prototyping the wearable device to better describe the vest and relation to the theoretical framework, from line 314-335.
Reviewer Comment:How many are the squares stitched on the t-shirt? Would a different shape (e.g., a rectangle, a circle, etc.) affect the performance of the device? Why did you choose the square shape for the sensing elements? You need to better clarify how form (size, shape) and function (pressure applied) correlate together. You need to better justify the reason why this is the final design.
Response:We added information about process and considerations during prototyping the wearable device to better describe the textile reflexes panels integration in the vest, line 314-335 and the working of the textile reflexes panel itself, line 230-236.
Reviewer Comment:Page 1, Line 29. You need to add few more references like some reviews that focus on the non-invasive monitoring of physiological parameters (e.g., in saliva, sweat, breath, etc.) using wearable devices.
Response:We considered the suggestions and added some references focusing on non-invasive monitoring.
Reviewer comment:Figure 2: The figure needs to have a higher resolution. The text in the boxes on the left hand side is not clear enough. You might also need to increase the size of the font you used.
Response: A higher resolution image has been added, we think that this has improved readability of the text in the figure.